# Anti-Mycobacterial Drug Resistance in Japan: How to Approach This Problem?

**DOI:** 10.3390/antibiotics11010019

**Published:** 2021-12-24

**Authors:** Keisuke Kamada, Satoshi Mitarai

**Affiliations:** Department of Mycobacterium Reference and Research, the Research Institute of Tuberculosis, Japan Anti-Tuberculosis Association, Tokyo 204-8533, Japan; keisukekmd@gmail.com

**Keywords:** drug resistance, tuberculosis, *Mycobacterium avium-intracellulare* complex, *Mycobacterium abscessus* species, Japan

## Abstract

Mycobacteriosis is mainly caused by two groups of species: Mycobacterium tuberculosis and non-tuberculosis mycobacteria (NTM). The pathogens cause not only respiratory infections, but also general diseases. The common problem in these pathogens as of today is drug resistance. Tuberculosis (TB) is a major public health concern. A major challenge in the treatment of TB is anti-mycobacterial drug resistance (AMR), including multidrug-resistant tuberculosis and extensively drug-resistant tuberculosis. Recently, the success rate of the treatment of drug-resistant tuberculosis (DR-TB) has improved significantly with the introduction of new and repurposed drugs, especially in industrialized countries such as Japan. However, long-term treatment and the adverse events associated with the treatment of DR-TB are still problematic. To solve these problems, optimal treatment regimens designed/tailor-made for each patient are necessary, regardless of the location in the world. In contrast to TB, NTM infections are environmentally oriented. *Mycobacterium avium-intracellulare* complex (MAC) and *Mycobacterium abscessus* species (MABS) are the major causes of NTM infections in Japan. These bacteria are naturally resistant to a wide variation of antimicrobial agents. Macrolides, represented by clarithromycin (CLR) and amikacin (AMK), show relatively good correlation with treatment success. However, the efficacies of potential drugs for the treatment of macrolide-resistant MAC and MABS are currently under evaluation. Thus, it is particularly difficult to construct an effective treatment regimen for macrolide-resistant MAC and MABS. AMR in NTM infections are rather serious in Japan, even when compared with challenges associated with DR-TB. Given the AMR problems in TB and NTM, the appropriate use of drugs based on accurate drug susceptibility testing and the development of new compounds/regimens that are strongly bactericidal in a short-time course will be highly expected.

## 1. Introduction

### 1.1. Epidemiological Reality of Mycobacterial Infections in Japan

#### 1.1.1. Tuberculosis

There have been four important changes in the epidemiology of tuberculosis (TB) in Japan since the 1940s. The first is the decline in the prevalence of TB. The incidence rate (per 100,000 population) of TB was over 100 in the 1960s; however, it quickly dropped to 10.1 in 2020 [1]. In 2020, the total number of TB notifications in Japan was 12,739. Japan is expected to become a low-incidence country (incidence <10) by 2021. The second change is the aging of patients with TB. Approximately 70% of patients with TB in Japan are aged over 65 years, and most of them developed TB due to the reactivation of latent infection [1]. The third change is the increasing number of patients in urban areas. This is particularly evident in the population younger than 39 years old [1]. The fourth is the increase in the number of foreign-born patients with TB. Although the number of foreign-born new patients with TB increased from 952 in 2010 to 1667 in 2018, fewer foreign-born patients with TB were reported in 2020, mainly due to coronavirus disease pandemic-related restrictions [1]. Moreover, although only 11.1% of all patients with TB are foreign-born, the proportion is increasing among young patients with TB. In particular, 71.3% of new patients with TB aged 20–29 years in 2019 are foreign born [1]. Given the relatively high DR-TB in Asian countries, the proportion of DR-TB in Japan is increasing again in these years. 

#### 1.1.2. Non-Tuberculous Mycobacteria

There are regional differences in the prevalence of non-tuberculous mycobacterial (NTM) pulmonary diseases (NTM-PD) and their pathogenic species. The incidence of NTM-PD in Japan is rapidly increasing. Namkoong et al. [2] and Morimoto et al. [3] reported an increase in the number of patients with NTM-PD in Japan. In a survey of dominant commercial laboratories, Morimoto et al. analyzed NTM species that fulfil the bacteriological diagnostic criteria for NTM-PD and reported that *Mycobacterium avium*, *Mycobacterium intracellulare*, *Mycobacterium kansasii*, and *Mycobacterium abscessus* species (MABS) are the most common pathogenic agents that cause NTM PD in Japan [4] (Figure 1). Kamada et al. surveyed rapidly growing mycobacteria (RGM) isolated from clinical specimens at a laboratory center and 45 hospitals and reported that MABS accounted for approximately 60% of all RGM identified, followed by *M. fortuitum* and *M. chelonae* [5]. Cowman et al. reported that with an incidence of 23–37 cases per 100,000, the annual prevalence of NTM-PD in Japan may be higher than that in the United States, which is 33–65 cases per 100,000 [6]. In addition, Hoefsloot et al. investigated more than 20,000 NTM-PD cases in 30 countries and reported that 31% of cases in Asia were caused by RGM including MABS, which is the highest proportion amongst the regions investigated [7]. Although epidemiological studies of NTM diseases are still few, the reported statistics indicate that Japan has a high incidence of NTM disease.

This is an estimate based on laboratory data aggregated from cases that meet bacteriological diagnostic criteria of NTM-PD issued by the American Thoracic Society/Infectious Diseases Society of America. Identification of NTM species was performed by DNA-DNA hybridization mycobacteria kit (Kyokuto Pharmaceutical Industrial Co., Ltd, Tokyo, Japan).

### 1.2. Current Treatment Strategies and Problems

#### 1.2.1. Tuberculosis

The standard short-course regimen is utilized for the initial treatment of TB in Japan. It is the administration of a combination of isoniazid (INH), rifampin (RIF), pyrazinamide (PZA), and ethambutol (EMB) or streptomycin (STR) for 2 months in an intensive phase, followed by a continuation phase that involves the administration of INH and RIF for 4–7 months, depending on the immunological status of the patient [8]. In Japan, although approximately 90% of new patients with TB are pan-susceptible, the treatment success for TB in 2019 was 66.3% [1]. The major reason for this is the presence of complications in older patients, who constitute the majority of Japanese patients with TB. Many TB deaths have been reported in this cohort of older patients (23.1% in 2019) [1]. Thus, even without serious drug resistance, treatment of new TB cases is a major problem in Japan.

The treatment of DR-TB has drastically changed in recent years due to the introduction of new and repurposed drugs. The list of recommended treatment regimens for DR-TB is presented in Table 1. In Japan, five drugs are preferred for the treatment of MDR (multidrug resistant)-TB: bedaquiline (BDQ), fluoroquinolones, delamanid, pyrazinamide, and D-cycloserine (DCS). If these drugs are not applicable, STR, kanamycin, enviomycin, ethionamide, or para-aminosalicylate are used. Drug selection is based on the reliable drug susceptibility testing (DST) results. Therefore, the Japanese national TB program employs a tailor-made strategy for the management of TB.

The World Health Organization (WHO) recommends all-oral treatment regimens without injectable drugs for the treatment of TB [10]. In addition, the 2020 revision of the guidelines recommend switching from injections to BDQ in short-term regimens [11]. In response to these changes, the definitions of pre-extensively drug resistance- and extensively drug resistance (XDR)-TB were changed in 2021 (Table 2) [14]. These recommended treatment regimens (Table 1) can be divided into two strategies: long-term and short-term regimens with all oral drugs. Long-term regimens require a minimum treatment period of approximately 18–20 months. Although shortening the duration of treatment is appealing [10], the patient must have been treated with a second-line drug for less than one month and be susceptible to fluoroquinolones to be eligible for the WHO shorter regimen [10].

BPaL is an innovative short treatment regimen for MDR-TB in which BDQ, pretomanid, and LZD are administered for 26 weeks (if the sample culture is positive at week 16, treatment extends to 39 weeks) [13]. In an open-label, single-group study, Conradie et al. reported that 90% of patients treated with BPaL had a favorable outcome [13]. Although BPaL can be used for the treatment of XDR-TB, peripheral neurological toxicity due to LZD occurs in more than 80% of patients [13]. A significant future challenge will be minimizing the incidence of adverse events caused by therapeutic agents. Other LZD analogues could be used as replacements for LZD.

Although shortening the duration of treatment for TB has been a longstanding issue, Dorman et al. recently reported in an open-label, randomized, controlled phase 3 trial that a short-term (four months) regimen for the treatment of newly diagnosed TB, in which RIF was replaced with rifapentine and moxifloxacin (MFLX), showed non-inferior results to a standard six-month regimen [15].

#### 1.2.2. *Mycobacterium avium-intracellulare* Complex

According to new ATS/ERS/ESCMID/IDSA guidelines, the standard treatment for macrolide-susceptible MAC-PD is the administration of a combination of a macrolide, RIF, and EMB [16]. Griffith et al. reported that bacteriological success in the treatment of macrolide-susceptible MAC-PD ranges from 71% to 85% [17]. Diel et al. reported that treatment of macrolide-susceptible MAC-PD for more than 12 months is associated with better results [18]. In addition, the ATS/ERS/ESCMID/IDSA guidelines recommend treatment for more than 12 months after negative culture conversion [16]. In their single-center retrospective studies, Kadota et al. and Furuuchi et al. reported that discontinuing treatment in less than 15 months after sputum culture conversion was a risk factor for relapse [19,20]. In addition, the benefit of adding RIF to the combination of macrolide and EMB for the treatment of macrolide-susceptible MAC-PD is still unclear. A phase III clinical trial in which these regimens are compared is currently in progress (NCT03672630). Since macrolide is the key drug for the treatment of MAC-PD, its interaction with RIF is crucial.

Given the major anti-mycobacterial effect of macrolides, acquired resistance to these drugs is a major clinical problem. Griffith et al. showed that the prognosis of patients with macrolide-resistant MAC-PD is poor (34% mortality after one year of treatment if sputum culture conversion was not achieved) and that long-term use of aminoglycosides (median, six months) in combination with surgical treatment is useful in achieving negative sputum culture conversion [21]. Furthermore, Morimoto et al. reported that additional use of a fluoroquinolone does not contribute to negative conversion of sputum culture in cases of macrolide-resistant MAC-PD [22].

Amikacin (AMK) is considered to be important in the treatment of refractory MAC-PD, including macrolide resistant MAC-PD. However, adverse events, such as hearing impairment and renal dysfunction, are associated with administration of the total dose of AMK. It was recently reported that administration of AMK liposomal inhalation suspension (ALIS) is a solution to this problem. Griffith et al. conducted an open-label, non-placebo-controlled, phase III trial to evaluate the efficacy of ALIS in patients with refractory MAC-PD who did not achieve negative sputum cultures after six months of treatment. The study showed that patients who added ALIS to guideline-based therapy had a higher rate (29%) of negative sputum culture conversion within six months than those who continued guideline-based therapy alone (8.9%) [23]. Although ALIS did not increase the incidence of hearing loss or renal dysfunction in the patient group compared to the control (standard treatment) group, mostly respiratory adverse events, such as dysphonia, cough and dyspnea, were observed in the patient group, especially within the first month; in addition, 17.5% of the patients could not continue taking ALIS [23]. Winthrop et al. also examined the effect of extending treatment with ALIS to 20 months in patients who did not achieve negative sputum cultures at six months (prior-ALIS Cohort) [24]. Although 14% of the patients in that study achieved negative sputum culture conversion, 26% temporarily discontinued ALIS due to adverse events such as dyspnea, whereas 8% permanently discontinued the treatment due to the occurrence of adverse events such as allergic alveolitis [24].

Clofazimine (CLO) is also a candidate for the treatment of refractory MAC-PD. In a retrospective chart review, Martiniano et al. reported the efficacy of regimens that contain CLO for the treatment of 26 cases of refractory MAC infections at two hospitals. The results of that study indicated that 42% of patients treated with CLO for at least six months achieved negative culture conversion within 12 months [25]. In a single-center prospective cohort study, Kwak et al. reported that eight patients with NTM-PD and a CLO minimum inhibitory concentration (MIC) ≤0.25 mg/L who were treated with a regimen that contained CLO achieved negative sputum culture conversion [26]. Currently, a phase II trial for the evaluation of the efficacy and safety of six-month CLO treatment for MAC-PD is in progress (NCT 04922554).

Rifabutin (RBT) and sitafloxacin (STX) may also be used to treat refractory MAC-PD. Some experts recommend adding high-dose RFB (300–600 mg/day) to the treatment regimen, especially for patients with macrolide-resistant MAC-PD and those who cannot use EMB; however, gastrointestinal symptoms and other adverse events may pose a problem [17,21]. In a retrospective, single-center, observational study, Asakura et al. reported that STX is effective as a salvage drug for refractory MAC-PD [27]. The efficacies of these drugs (CLO, RBT, and STX) for the treatment of MAC need to be evaluated further.

#### 1.2.3. *Mycobacterium abscessus* Species

MABS is naturally resistant to many antimicrobial agents and is one of the most refractory mycobacterial species. The ATS/ERS/ESCMID/IDSA guidelines recommend combination chemotherapy aimed at synergies [16]. If the MABS is susceptible to macrolides, at least three susceptible antimicrobial agents, including macrolides, should be used in the induction phase of the treatment regimen. In the maintenance phase, the use of two or more susceptible drugs in all cases is recommended [16]. However, it is often difficult to select appropriate drugs and to keep sputum culture negative in cases of infection caused by macrolide-resistant MABS. In their study, Pasipanodya et al. reported that 54% of patients with infections caused by *Mycobacterium abscessus* subsp. *massiliense* (MMA), which is often macrolide-susceptible, sustained negative sputum conversion, whereas only 34% of those infected by *Mycobacterium abscessus* subsp. *abscessus* (MAB), which is often macrolide-resistant due to *erm*(41), achieved negative conversion [28].

If MABS is macrolide-resistant, treatment should be started with at least four other susceptible drugs. However, the efficacy of therapeutic agents other than macrolides for MABS as recommended in the ATS/ERS/ESCMID/IDSA guidelines [16] is unclear. Kwak et al. reviewed the efficacies of injectable drugs and reported that imipenem (IPM) for MABS-PD and AMK for MAB-PD are associated with treatment success [29]. Chen et al. also analyzed the treatment regimens of 244 patients with MABS-PD in a single-center retrospective study, and reported that the inclusion of AMK, IPM, LZD, and TGC in the treatment regimen was associated with treatment success [30]. Conversely, Park et al. reported that the choice of AMK and β-lactam antibiotic injection based on MIC is not correlated with successful treatment of MABS-PD [31]. Martiniano et al. conducted a retrospective chart review to evaluate the efficacy of regimens that contain CLO for 36 patients with MABS in two hospitals and reported that use of CLO for at least 6 months achieved 50% negative culture conversion within 12 months [25]. A study designed to determine the Optimal Regimen for *Mycobacterium abscessus* Treatment (FORMaT) is currently in progress (phase III, NCT04310930).

## 2. Drug Susceptibility Testing and Limitations

### 2.1. Principle of Drug Susceptibility Testing: Why This Process Is Necessary

Drug susceptibility testing (DST) is essential for the selection of appropriate therapeutic agents for treating bacterial infections. Performing DST prior to the initiation of treatment for mycobacterial infections is recommended [32]. DST is important, especially in cases of high antimicrobial resistance. The optimal treatment for each patient is different, as the antibiotic effects of drugs on each pathogen vary. It is essential to quickly obtain accurate DST results to enable timely initiation of appropriate treatment.

### 2.2. Mycobacterium tuberculosis

#### 2.2.1. Phenotypic Drug Susceptibility Testing

Phenotypic drug susceptibility testing (pDST) is fundamental for estimating the clinical outcomes of chemotherapy. The correlation between chemotherapy and clinical efficacy has been evaluated. Some standard DST methods have been established for *Mycobacterium tuberculosis* (Mtb) infections. However, the biggest disadvantage of pDST for Mtb is its time-consuming nature. It is necessary to use pure Mtb culture for testing; therefore, the procedure takes time. If NTM coexists in the specimen, it may be difficult and time-consuming to separate the Mtb bacteria from the NTM [33]. Given the standard treatment regimen for TB, patients with drug-resistant TB may be treated with an inappropriate regimen until treatment failure is noted or the test results are confirmed. There are two additional disadvantages of the pDST. One is the need for appropriate human resources with reliable bacteriological analysis skills. The other is the need for a secure biosafety facility, which is often difficult to access, especially in settings where resources are limited.

There has been another technical problem with pDST in recent years. According to the new WHO definition of XDR-TB, pDST for LVX, MFX, BDQ, and LZD is necessary; however, the commercial MGIT DST kit is not available from Becton Dickinson as of November 2021, even in industrialized countries. This is a major obstacle to acquire routine DST for these drugs. Farooq et al. reported that in 2019, only approximately 30% and 60% of 28 reference laboratories in Europe were able to perform pDST of BDQ and LZD, respectively [34].

#### 2.2.2. Genotypic Drug Susceptibility Testing

Genotypic drug susceptibility testing (gDST) is the estimation of pDST results from genetic variants, mutations, insertions, and deletions in drug resistance genes, and is expected to solve the problems of pDST described above. However, to improve the validity of gDST results, it is necessary to obtain and accumulate high-quality pDST results. Unfortunately, there is no systematic data collection mechanism for this as of 2021.

gDST is widely used in clinical practice for the diagnosis of resistant TB. Approximately 95% of RIF drug-resistance mutations in Mtb occur in the 81 bp rifampicin resistance-determining region of the *rpoB* [35]. This mutation can be detected directly in clinical specimens in less than two hours using Xpert MTB/RIF. In 2018, the Global Laboratory Initiative (GLI) recommended Xpert MTB/RIF as a screening tool in its algorithm for deciding on a DR-TB treatment regimen. Confirmation of RIF resistance using Xpert MTB/RIF should be followed by a line probe assay to confirm fluoroquinolone (and second-line injectable drug) resistance [36]. This strategy is now widely used in many countries where M/XDR-TB is prevalent.

Many studies on gDST have been conducted. In a systematic review, Kadura et al. reported the genetic mutations related to resistance to BDQ, LZD, CLO, and DMD [37]. However, the limited number of resistant isolates makes the validation between gDST and pDST difficult. An amplicon deep sequencing gDST assay based on multiplex PCR (Deeplex-MycTB, Genoscreen, Lille, France) was recently developed. Deeplex-MycTB is an all-in-one targeted deep-sequencing assay of a 24-plexed amplicon mix that covers 18 gene regions. Feuerriegel et al. reported the simultaneous analysis of resistance gene mutations against 14 antimicrobial agents using Deeplex-MycTB. The results of the study showed that pDST has good concordance with first-line drugs and fluoroquinolones [38]. Whole-genome sequencing is a promising method that can cover all potential mutations/indels. Smith et al. reported that MinION (Oxford Nanopore, Cambridge, UK) may be an alternative method to next-generation sequencing methods in the genetic analysis of Mtb [39]. It is necessary to establish a universal system to collect clinical isolates resistant to these drugs and to evaluate the correlation between pDST results and mutations/indels.

### 2.3. Slowly Growing Mycobacteria (MAC)

The details of the DST of *Mycobacterium avium*–*intracellulare* complex (MAC) with MIC breakpoints for five antimicrobial agents (Table 3) are indicated in Clinical Laboratory Standards Institute (CLSI) documents M24 and M62 [40]. MIC is the only clinically feasible parameter used to evaluate the efficacy of anti-microbial agents against MAC, mainly due to their variable susceptibility to the same drug. Except for macrolides and AMK in MAC infections, no clear association between DST results and clinical efficacy has been confirmed. MIC is more reliable than the breakpoint test because of its quantitative nature.

Although gDST is not often used for MAC infections, genetic mutations related to resistance to macrolides and AMK have been reported. Griffith et al. and Moon et al. showed that point mutations (A2058 or A2059) in the 23S rRNA (*rrl*) are present in more than 90% of macrolide-resistant MAC [21,41]. Since this region is the target point of macrolide therapy, the mutation reduces affinity to macrolide, leading to resistance. Brown-Elliott et al. reported that AMK resistance in MAC is caused by mutations in the 16S rRNA (*rrs*) and linked to an MIC > 64 μg/mL [42]. gDST will be useful for timely care in the follow-up of patients with refractory MAC infection.

### 2.4. Rapidly Growing Mycobacteria (MABS)

Among RGM, only DST of MABS is clearly associated with clinical efficacy. In CLSI M62, MIC breakpoints for susceptibility interpretation are indicated for 11 antimicrobial agents (Table 3) [40]. In the new ATS/ERS/ESCMID/IDSA guidelines, AZM is recommended as the first choice of macrolide compared to CLR. In addition, AZM is not included in the MIC susceptibility interpretation table. Furthermore, the CLSI M62 indicates that CLR is the class drug for macrolides and the only macrolide that needs to be tested. The MIC test for AZM is not reliable; thus, its clinical efficacy should be estimated using susceptibility to CLR. However, a direct method of determining MIC is necessary, even for AZM, to secure valuable MIC information for the treatment of MABS infections.

Macrolide resistance in MABS is the most investigated area of the gDST of RGM. Two major genes have been identified thus far. The first gene is *erm*(41), which is responsible for macrolide-induced resistance. Exposure to macrolides activates the expression of the *erm*(41) gene, and the product methylates the target, resulting in a lower affinity for macrolide. Resistance induced by *erm*(41) accounts for the majority of macrolide resistance mechanisms in MAB and *M.abscessus* subsp. *bolletii* (MBO), whereas *erm*(41) is truncated in *Mycobacterium abscessus subsp. massiliense* (MMA) [43]. T28C variants have been reported to lose *erm*(41) function and show macrolide susceptibility. Brown-Elliot et al. reported that determination of the *erm*(41) gene sequence in MAB is useful for predicting macrolide susceptibility [44]. In addition, sequencing is recommended in the CLSI guidelines [32].

Another gene related to macrolide resistance is the *rrl* (23S rRNA) gene, as is the case with MAC. Wallace reported that mutation of the *rrl* gene in MABS also confers acquired resistance to macrolide [45]. MMA and MAB with *erm*(41) T28C sequevar are also known to acquire resistance to macrolide by *rrl* mutation. In addition, Mougari et al. and Kim et al. reported that AMK MIC > 64 μg/mL is linked to *rrs* mutations in MABS [46,47].

## 3. Drug Resistant Status of Major Mycobacterial Pathogens

### 3.1. M. tuberculosis

#### 3.1.1. Characteristics of Drug-Resistant Tuberculosis in Japan (Drug Resistance, Prognosis)

In Japan, among 5209 newly registered patients with pulmonary TB with confirmed DST results in 2020, 297 (5.7%) had INH resistance, 60 (1.2%) had RIF resistance, and 46 (0.9%) had MDR [1]. Of the 4624 native patients with DST results, 227 (4.9%) had INH resistance, 33 (0.7%) had RIF resistance, and 23 (0.5%) had MDR [1]. In the last decade, there have been approximately 50 new cases of MDR-TB per year in Japan, with no significant change. Regarding STR and EMB, 6.8% of new patients and 12.2% of retreated patients were resistant to STR, and whereas 1.3% of new patients and 4.5% of retreated patients were resistant to EMB [1]. Since the DST results of other antibiotics are not registered, the exact proportion of patients with resistance to them is unknown.

The proportions of new patients with both INH resistance and MDR tends to decrease with increasing age. In 2020, the proportion of retreated patients with INH resistance aged 40–79 years was approximately 20%, which was higher than that of new patients (approximately 5%) [1]. A higher percentage of foreign-born patients have resistant TB than native patients. In 2020, among 530 patients with DST results, 68 (12.8%) had INH resistance, 27 (5.1%) had RIF resistance, and 23 (4.3%) had MDR. Only 15 patients were retreated, whereas half of them had MDR [1]. In a nationwide survey conducted by Ryoken consortium in 2012–2013, resistance to levofloxacin (LVX) was observed in 3.2% of new patients and 6.1% of retreated patients [48]. There is insufficient information on DR-TB in patients living with human immunodeficiency virus (HIV). Of the new TB cases in 2020, HIV test results were known for only 877 cases (6.9% of the total), of which only 31 cases were positive for HIV [1]. Among the 31 cases, drug susceptibility results were known for 16 cases and INH resistance was observed in only 1 case (6.3%) (unpublished data). These indicate that anti-TB drug resistance in Japan is not a serious issue in new or retreatment cases.

There are limited reports on the treatment outcomes of MDR-TB in Japan. Kawatsu et al. analyzed the outcomes of 172 patients with MDR-TB using surveillance data collated from 2011 to 2013 in Japan [49]. The treatment completion rate was 71.6% in the 15–64 years age group, and 39% in those over 65 years old. The mortality rate was 4.2% in patients aged 15–64 years, and 49.4% in those older than 65 years [49]. Old age is a major factor for treatment failure, mainly because of the complications older patients have, regardless of anti-TB drug susceptibility.

It is not easy to diagnose MDR-TB in Japan, even using gDST. This is inevitable because of the small number of new cases in Japan. Xpert MTB/RIF is used for gDST; however, the low prerequisite probability of RIF-resistant TB reduces the positive predictive value of the test. Even if RIF resistance is confirmed, the probability of detecting MDR-TB is approximately 60–80% [48,50]. Thus, it is necessary to confirm INH susceptibility.

In 2021, the Ministry of Health, Labour, and Welfare revised the standards for TB treatment in Japan. This revision indicates that the principle of MDR-TB treatment in Japan is to select five drugs from the presented list (Table 4) and to continue treatment for 18 months after negative sputum culture conversion. LVX and BDQ should be used preferentially, and the remaining drugs should be selected from the other drugs. The omission of LZD and CLO from this list is a significant issue. It is difficult to choose a treatment regimen for pre-XDR-TB and XDR-TB without LZD and CLO. In fact, they are often used for the treatment of pre-XDR-TB and XDR-TB in Japan. In contrast, there is a screening system for the use of BDQ and DMD by pulmonologists. Its aim is to ensure the appropriate use of new anti-TB drugs to prevent the emergence of resistance. This system must be adaptable to LZD and CLO, which are already widely available in the market for other purposes. In addition, the use of DMD is approved in Japan, but pretomanid (PTO), a drug of the same class, is not approved; therefore, the BPaL regimen cannot be selected.

In Japan, because the aforementioned drugs cannot be used (or are difficult to use) in the treatment of MDR-TB, DMD tends to be used more widely than the international recommendations (Table 1), but its validity needs to be further evaluated.

#### 3.1.2. Treatment Regimen: Tailor-Made Treatment

In Japan, a tailor-made regimen is utilized depending on the DST results of the patient, especially for retreatment cases. In addition to rapid and appropriate selection (gDST) of a treatment regimen, it is ideal to determine the optimal drug dosage and the duration of treatment through therapeutic drug monitoring (TDM) and biomarker screening. New technologies, such as liquid chromatography tandem mass spectrometry, may help in the TDM of many therapeutic agents.

The optimal dosage should be determined according to the pharmacokinetic (PK) and pharmacodynamic (PD) parameters of the host and the MIC of the drug against MTB. In particular, the dosage and appropriate blood concentration of LZD that causes serious adverse events remains controversial. In a systematic review, Lifan et al. showed that reduction of LZD dosage to ≤600 mg/day reduced adverse events such as myelosuppression and gastrointestinal disorders but not neuropathy, maintaining the success rate of the treatment [51]. However, how much LZD can be reduced while maintaining efficacy when adverse events occur is unclear. To solve this problem, more data on the correlation between PK/PD/TDM, therapeutic effects, and adverse effects are needed. Ideally, relevant information on appropriate indicators should be available prior to initiation of therapy to ensure that effective drug combinations are used. As there are several drug combinations used in the treatment of M/XDR-TB, it may be necessary to know the effective indicators to ensure that the best combination is used.

Monitoring treatment effect is also an important issue. Conventional culture examination is recommended by the WHO; however, it is time consuming. Therefore, it is important to establish biomarkers that can be used to evaluate the response to treatment without the need for culture [52]. Kawasaki et al. reported the usefulness of lipoarabinomannan in sputum as a marker that reflects the bacterial load of Mtb [53]. In a prospective proof-of-concept study, Sakashita et al. reported that MPT-64 in sputum is useful, not only for diagnosis, but for the prediction of culture results during treatment as well [54]. It is necessary to combine treatment regimens and biomarkers in future clinical trials to evaluate their correlation with treatment success [52].

### 3.2. Mycobacterium avium-intracellulare Complex

#### 3.2.1. Characteristics of Pulmonary *Mycobacterium avium-intracellulare* Complex Disease in Japan (Drug Resistance, Prognosis)

It is difficult to describe the exact prognosis of MAC-PD in Japan, mainly because of lack of systematic statistics. There are several reports on the prognostic factors of MAC-PD. Hayashi et al. analyzed 634 patients with MAC-PD and reported significantly poor prognosis in patients with FC/FC-NB compared to others [55]. In their analysis of the vital statistics of Japan, Morimoto et al. reported an annual mortality rate of 1–2% for patients with NTM infections [3]. Single-center retrospective studies conducted in Japan have shown that the presence of pulmonary cavities and macrolide resistance are the main factors that worsen the prognosis of MAC-PD [22,56,57]. Gochi et al. investigated the prognosis of 782 MAC-PD cases and reported that the five-year and 10-year mortality rates due to progression of MAC-PD were 2% and 4.8%, which increased significantly to 9% and 25% in patients with cavity [56]. Kadota et al. reported the outcomes of 33 cases of macrolide-resistant MAC-PD; 75% of the patients had cavities at the start of treatment and the mortality rate at one year was 6% [57]. Morimoto et al. analyzed 102 patients with macrolide-resistant MAC-PD and reported that 66% had cavities. The five-year mortality rate in that study was 29%, which is not significantly different from the mortality rate for MDR-TB at the time the study was conducted [22].

#### 3.2.2. Drug Resistance of *Mycobacterium avium-intracellulare* Complex

There are no data on the proportion of patients with macrolide-resistant MAC in Japan. Morimoto et al. suggested that the causes of macrolide resistance include macrolide monotherapy, macrolide plus fluoroquinolone therapy, and treatment without EMB [22]. In addition, Kadota et al. and Ito et al. showed that CLR + EMB treatment regimens without RIF do not increase macrolide resistance [57,58]. In their analysis of Japanese national health insurance receipt data, Iwao et al. reported that the percentage of patients who received CLR monotherapy for MAC-PD for more than 3 months was 9.2% [59]. Morimoto et al. reported that among 77 patients with macrolide-resistant MAC-PD, approximately 20% (15 cases) exhibited AMK MIC >32 μg/mL [22]. Yamaba et al. performed MIC tests on a total of 189 MAC isolates and reported that approximately one-fourth of the strains showed intermediate resistance or resistance to moxifloxacin. However, no mutations were identified in *the gyrA* and *gyrB* genes [60].

#### 3.2.3. Current Slowly Growing Mycobacteria (MAC) Drug Susceptibility Testing Problems in Japan

The biggest problem with DST of MAC (slowly growing mycobacteria [SGM]) in Japan is the culture medium. CLSI has already recommended DST with cation-adjusted Mueller–Hinton Broth (CAMHB) instead of Middlebrook 7H9 Broth. However, in Japan, only BrothMIC NTM (Kyokuto Pharmaceutical Industrial Co., Ltd., Tokyo, Japan) using Middlebrook 7H9 Broth is available for DST, which means no evidence-based clinical interpretation of MIC data.

Woods et al. reported compatibility between CAMHB and 7H9 media in macrolide DST [61]. However, AMK compatibility has not been reported in literature; therefore, AMK resistance may not have been accurately assessed in Japan.

### 3.3. Mycobacterium abscessus Species

#### 3.3.1. Characteristics of *Mycobacterium abscessus* Species Pulmonary Disease in Japan (Drug Resistance, Prognosis)

MABS-PD infections are difficult to treat, even when compared to MAC-PD. Morimoto et al. investigated the prognosis of 121 patients with MABS-PDs in a multicenter retrospective study conducted in Japan [62]. The median observation period in that study was 38 months, and sputum culture conversion after treatment was significantly higher in patients with MMA (72%) than in those with MAB (35%) [62]. In addition, the incidence of all-cause death during the observation period was predominantly higher in the MAB group (22%) than in the MMA group (3%) [62]. In a single-center retrospective observational study, Fujiwara et al. reported age, history of NTM-PD, and cavitary lesions as clinical risk factors associated with treatment failure in MABS-PD, and showed that advanced age, cavitary lesions, history of NTM lung disease may be clinically important factors related to unfavorable treatment outcomes [63].

Aono et al. and Kamada et al. reported the MIC data for MABS clinical isolates in Japan. In their studies, the proportions of MAB isolates with the *erm*(41) T28C variant were approximately 10% [64,65]. Approximately 3% of MMA isolates showed macrolide-acquired resistance due to *rrl* mutations [64,65]. The proportion of cases of AMK resistance in MABS was low, ranging from 1% to 5%. However, there were no significant differences between these proportions and those reported for other countries [44,46].

#### 3.3.2. Current Rapidly Growing Mycobacteria Drug Susceptibility Testing Problems in Japan

In Japan, the difference between DST for RGM and SGM is not well understood, and even now, MIC kit for SGM (BrothMIC NTM: Kyokuto Pharmaceutical Industrial Co., Ltd., Tokyo, Japan) is often used for the DST of RGM isolates [66,67] (Table 3). In addition, the capacity for DST of RGM is still limited. MIC tests are technically demanding in general, and MIC for RGM requires experienced technologists. It is expected that MIC can be automatically determined, and the time required to obtain MIC results can be shortened.

## 4. New Antimicrobial Therapeutic Candidates

Compared to other countries, it is a problem that there are several drugs not approved in Japan for mycobacterial infection, i.e., pretomanid, tigecycline, omadacycline, rifapentine, stezolid (SZD). However, approval of these drugs is not a fundamental solution to the problem. This part will focus on potential drugs and provide information to promote new research.

### 4.1. M. tuberculosis

#### 4.1.1. Oxazolidinones: Stezolid

It is known that oxazolidine inhibits protein synthesis by acting on the ribosomal S50 of bacteria, and that LZD is a typical agent [68]. Cynamon et al. reported that SZD showed better anti-tuberculosis activity than LZD in a mouse model of tuberculosis [69]. Williams et al. reported the possibility of shortening the duration of treatment if SZD is added to first line TB drugs [70]. In addition, Wallis et al. reported that SZD administered for 14 days to treatment-naïve patients with TB demonstrated anti-tuberculosis activity and good tolerability [71].

#### 4.1.2. DprE1 Inhibitors: PBTZ169, OPC-167832, TBA-7371

Decaprenylphosphoryl-β-D-ribose 2-epimerase (DprE1) is an essential enzyme needed for cell wall synthesis in Mtb and is responsible for the formation of lipoarabinomannan and arabinogalactan [68]. Since covalent or non-covalent binding of ligands to DprE1 leads to inactivation of DprE1 and death of Mtb, several drugs that target DprE1 have been developed. Robertson et al. reported that of these drugs, OPC-167832 showed superior efficacy in a mouse model of TB [72]. In addition, Hariguchi et al. reported that OPC-167832 showed a significant combination effect with DMD, BDQ, and LVX in a murine model of TB. It also showed a better bacterial load reduction and relapse-prevention effect than in standard therapy in a 3–4 drug combination with DMD, BDQ, moxifloxacin (MXF), and LZD [73].

#### 4.1.3. Diarylquinolines: TBAJ-587, TBAJ-876

Diarylquinolines exhibit antimicrobial activity by inhibiting the ATP synthase of mycobacteria. BDQ is the most representative diarylquinoline [68]. BDQ is a lipophilic drug that is metabolized by cytochrome P450 and accumulates in tissues for a long time as a result of its interaction with intracellular phospholipids [68]. An important adverse effect of BDQ is prolongation of the QT interval due to potent inhibition of hERG, a potassium channel protein in the heart [74].

Recently, new diarylquinolines (TBAJ-587, TBAJ-876) with different chemical characteristics from BDQ have been developed. Xu et al. reported that TBAJ-587 showed better anti-TB activity than BDQ in a mouse model of TB [75]. In that study, TBAJ-587 was shown to be effective against Mtb with *Rv0678* mutation, which overexpresses the MmpL5/MmpS5 efflux transporter, resulting in BDQ resistance [75]. Furthermore, Almeida et al. reported that TBAJ-876 is more effective than BDQ against the *Rv0678* mutant in a mouse model [76]. In addition, Sutherland et al. reported that TBAJ-876 has less hERG channel inhibition effects and is expected to reduce QT prolongation [74].

### 4.2. M. avium-intracellulare Complex

#### 4.2.1. Bedaquiline

BDQ is also expected to be a salvage drug for MAC infections. Kim et al. reported that, in vitro, more than 90% of the strains of *M. avium* and *M. intracellulare* have an MIC of BDQ < 0.016 μg/mL, even among macrolide-resistant strains [77]. Philley et al. reported that four out of six patients with refractory MAC-PD treated with additional BDQ achieved at least one negative sputum culture within six months [78]. Alexander et al. reported relapse in seven of 13 cases of *M. intracellulare* PD treated with BDQ. The *M. intracellulare* isolates from the relapsed patients showed mutations in the *mmpT5* gene, which is similar to the *mmpR5* gene involved in the drug efflux that causes BDQ resistance in Mtb. In addition, the *mmpT5* mutation caused a two- to eight-fold elevation in the MIC of BDQ and CLO [79].

#### 4.2.2. Benzimidazoles: SPR719

DNA gyrase is composed of two subunits, *gyrA* and *gyrB*. *gyrA*, the catalytic subunit, is the major target of fluoroquinolones, whereas *gyrB*, the ATPase subunit, is a minor target [80]. Locher et al. reported that SPR719 exhibits antibacterial activity by inhibiting *gyrB* in vitro against a wide range of mycobacterial species [80]. Pennings et al. reported that, in vitro, SPR719 in combination with EMB has a prolonged bactericidal effect on MAC and inhibits the emergence of SPR719-resistant strains [81].

### 4.3. M. abscessus Species

#### Omadacycline

Omadacycline (OMC) is an aminomethylcycline and a tetracycline class antimicrobial. It works against bacterial ribosomes and inhibits protein synthesis [82]. Bax et al. reported that OMC has good antibacterial activity against MABS, similar to tigecycline in vitro [83]. Gotfried et al. reported that OMC is associated with fewer gastrointestinal adverse events, such as vomiting, compared with tigecycline [84]. Morrisette et al. reported that of 12 patients with MABS, including non-pulmonary infections, who were treated with regimens that contained OMC for a median of 6.2 months, nine had good outcomes [85].

## 5. Conclusions

The incidence of TB in Japan is constantly decreasing, and that of DR-TB is still at a low level. However, vulnerable elderly people will become the majority of patients with TB in the near future, as in the United states and European countries. Therefore, it is necessary to develop appropriate strategies and methodologies for the implementation of personalized treatments.

NTM infections are becoming a serious health challenge in Japan because of the lack of rapid and accurate diagnostic methods, adequate DST, and effective therapeutic agents. The prevalence of NTM-PD in Japan is high. In many settings, the incidence of NTM disease increases with decreasing incidence of TB. Thus, there is a need to establish a regionwide surveillance system for NTM diseases. This will help in the development of strategies against the spread of NTM diseases.

## Figures and Tables

**Figure 1 antibiotics-11-00019-f001:**
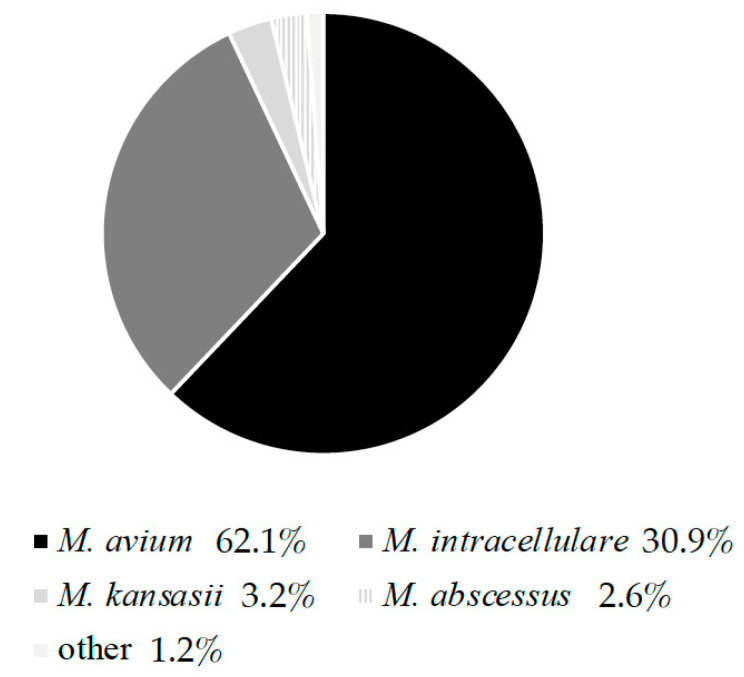
Prevalence of NTM-PD in Japan. Prepared based on Ref [4].

**Table 1 antibiotics-11-00019-t001:** Recommended treatment regimen for MDR TB. Prepared based on Refs. [9,10,11,12,13].

Regimen	ATS/CDC/ERS/IDSA	WHO Longer Regimen	WHO Shorter Regimen *	BPaL	JSTB ^a^
Year	2019	2020	2020	2020	2020
Number of drugs					
Intensive phase	5	4	7	3	5
Consolidation phase	4	3	4		
Treatment duration					
Intensive phase	5–7months **	6 months or longer **	4–6months		
Total	15–21months **^b^	15–17months **	9–12months	6–9months	18months **
Drugs	Strongly: LVX ^c^ (MFLX ^d^), BDQ Conditional: LZD ^e^, CLO ^f^, DCS, AMK ^g^, STR, EMB, PZA, carbapenem ^h^ DMD	Group A: LVX ^c^ (MFLX ^d^), BDQ, LZD ^e^, Group B: CLO ^f^, DCS Group C: EMB, DMD, PZA, carbapenem ^h^, AMK ^g^, ETO, PTO ^i^, PAS	Intensive phase: BDQ, INH, EMB, PZA, MFLX ^d^ (LVX ^c^) ETO, CLO ^f^ Consolidation phase: EMB, PZA MFLX ^d^, CLO ^f^	BDQ PTM LZD	Preferred: LVX ^c^, BDQ Second to the preferred: LZD ^e^ Additional: EMB, PZA, DMD CLO ^f^, DCS Conditional: STR, KAN ^j^, EVM ETO, PAS carbapenem ^h^

^a^ The Japanese Society for TB and NTM, ^b^ If pre-XDR or XDR, 15–24 months after culture conversion, ^c^ levofloxacin, ^d^ moxifloxacin, ^e^ linezolid, ^f^ clofazimine, ^g^ amikacin, ^h^ with clavulanic acid, ^i^ pretomanid, ^j^ kanamycin. * Must be susceptible to fluoroquinolones, and previous exposure of less than 1 month duration to the second-line drugs. ** After culture negative conversion.

**Table 2 antibiotics-11-00019-t002:** Changes in the new WHO definition for the category of drug resistant TB. Prepared based on Ref. [14].

Category	Privious Definition 2006–2021	New Definition 2021–
MDR	INH and RIF	INH and RIF
Pre-XDR	MDR + (any fluoroquinolone or at least one of three drugs *	MDR/RR + any fluoroquinolone
XDR	MDR + (any fluoroquinolone + at least one of three drugs *	MDR/RR + (any fluoroquinolone + at least one additional drug **

* second-line injectable drugs: capreomycin, KAN, AMK. ** group A drugs: BDQ, LZD.

**Table 3 antibiotics-11-00019-t003:** Difference in DST between MAC and MABS. Prepared based on Ref. [32,40].

Pathogen	MAC (Slowly Growing Mycobacteria)			MABS (Rapidly Growing Mycobacteria)		
Medium	CAMHB *			CAMHB		
Supplement	5% OADC			none		
Time to judge DST results	7 days			14 days **		
	MIC, μg/mL			MIC, μg/mL		
	S	I	R	S	I	R
CLR	≤8	16	≥32	≤2	7	≥8
AMK (IV)	≤16	32	≥64	≤16	32	≥64
AMK (liposomal inhaled)	≤64	-	≥128			
MFLX	≤1	2	≥4	≤1	2	≥4
LZD	≤8	16	≥32	≤8	16	≥32
IPM				≤4	8–16	≥32
MEPM				≤4	8–16	≥32
FOX ^a^				≤16	32–64	≥128
CIP ^b^				≤1	2	≥4
DOX ^c^				≤1	2–4	≥8
SXT ^d^				≤2/38	-	≥4/76

^a^ cefoxitin, ^b^ ciprofloxacin, ^c^ doxycycline, ^d^ trimethoprim-sulfamethoxazole. * In Japan, Middlebrook 7H9 broth is still used. ** To judge macrolide inducible resistance.

**Table 4 antibiotics-11-00019-t004:** Drugs approved by the Ministry of Health, Labour and Welfare for the treatment of MDR TB in Japan.

Priority	Drugs
Most preferred	LVX, BDQ
Additional *	EMB, PZA, DMD, DCS
Additional **	STR, KAN, EVMETO, PAS

Additional drugs * are used in preference to additional drugs **.

## Data Availability

All data are applicable in the paper.

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
