# Peer review of "Anti-Mycobacterial Drug Resistance in Japan: How to Approach This Problem?"

_antibiotics, 2021, doi:10.3390/antibiotics11010019_

Round 1

Reviewer 1 Report

The manuscript analyzes the epidemiological situation of infections by Mycobacterium spp. in Japan, the different strategies in the treatment of tuberculosis and other mycobacterial infections, the importance of phenotypic and genotypic drug susceptibility testing and their limitations, and finally, it briefly describes some of the new antibiotics under study. It is a very interesting, widely descriptive and well-organized work, which collects fundamental aspects of infection by these bacteria, which is why I recommend its publication in present form. 

Author Response

Response to reviewer #1:

Reviewer1: The manuscript analyzes the epidemiological situation of infections by Mycobacterium spp. in Japan, the different strategies in the treatment of tuberculosis and other mycobacterial infections, the importance of phenotypic and genotypic drug susceptibility testing and their limitations, and finally, it briefly describes some of the new antibiotics under study. It is a very interesting, widely descriptive and well-organized work, which collects fundamental aspects of infection by these bacteria, which is why I recommend its publication in present form. 

Response:

We thank you for the critical review of our manuscript. We are glad that you understand the significance of this work.

Reviewer 2 Report

This is an interesting review on the epidemiology and current therapeutic guidelines of Mycobacterium tuberculosis infection and non-tuberculous Mycobacterial infections in Japan. The paper is well written and all the necessary references have been cited. Minor comments to be addressed by the authors:

(a) It would be interesting to add more information about the epidemiology of DR-TB and XDR-TB in Japan focusing on specific groups such as immunocompromised hosts (HIV + patients for example) - if any data are available. 

(b) A few more info about NTMs especially their prevalence by adding a Table or a chart would be also useful.

(c) A list with all the abbreviations used in the text for the reader to understand which drug are the authors are refferring to (maybe at the top or the bottom of the manuscript).

Author Response

Response to reviewer #2:
Reviewer2: This is an interesting review on the epidemiology and current therapeutic guidelines of Mycobacterium tuberculosis infection and non-tuberculous Mycobacterial infections in Japan. The paper is well written and all the necessary references have been cited. Minor comments to be addressed by the authors:

Response:

We thank you for reviewing our manuscript and for providing insightful comments, which have greatly improved the quality of our manuscript. Please find our point-by-point responses to your comments below. The revisions are highlighted in yellow in the revised manuscript.

(a) It would be interesting to add more information about the epidemiology of DR-TB and XDR-TB in Japan focusing on specific groups such as immunocompromised hosts (HIV + patients for example) - if any data are available. 

Response:

Thank you for your comment. Because this is a very important point, we have added the necessary information to the revised manuscript, lines 405–409.

(b) A few more info about NTMs especially their prevalence by adding a Table or a chart would be also useful.

Response:

Thank you for your suggestion. We have added Figure 1, which shows the prevalence of NTM-PD in Japan.

(c) A list with all the abbreviations used in the text for the reader to understand which drug are the authors are referring to (maybe at the top or the bottom of the manuscript).

Response:

We have added a list of abbreviations (lines 43–94).

Reviewer 3 Report

This manuscript emphasises the importance of addressing the problem of AMR, in context with the latest advances in anti-TB therapies in Japan. Although very relevant to the current anti-TB agenda, this manuscript may benefit from the topics I leave below:

  • The title is quite informal, particularly following the use of "sort out". Authors to consider changing the title.
  • The abstract does not specify the aims of the review, why it brings novelty to the current literature, how the review is organised and conclusions. Authors to consider reviewing the abstract so key information is added.
  • Chapter 1, the introduction, has little information on what is happening in Japan. If the centre of the review is the anti-TB therapies in Japan, however there is not enough introductory facts from Japan that makes important to review and discuss data. Authors to consider adding more data from Japan, in order to accurately introduce the review.
  • Chapter 2, on drug susceptibility testing and limitations. Again, no data from its use in Japan. Authors to consider adding more data from Japan, in order to accurately address the reality and challenging of all the techniques in Japan. In other words, it would have to be clear why the authors develop this chapter so well in a review around TB in Japan.
  • Chapter 3 should be somehow linked with the chapter 1 and 2. In my opinion, it would be interesting to start with overall facts prior to a a focus on Japan, for all the subchapters. Authors to consider rewriting the review, including facts about Japan throughout the paper, and as an isolated chapter. It makes the overall manuscript more interesting for the reader, as facts between countries, for example, can be compared. The authors can provide factual comparisons.
  • Subchapter 3.3.2. - Are the authors able to provide references for this paragraph?
  • Chapter 4 - This is a chapter that could be generalised for all countries and not just Japan. In my opinion it this chapter would be interesting if the authors would compare drugs not available in Japan but available/recommended by the FDA, EMA, NICE/NHS, WHO, in order to "present" or suggest the future that Japan can have.
  • Chapter 4 - Can the authors justify why pretomanid is not in this list?

Author Response

Response to reviewer #3:
Reviewer3: This manuscript emphasises the importance of addressing the problem of AMR, in context with the latest advances in anti-TB therapies in Japan. Although very relevant to the current anti-TB agenda, this manuscript may benefit from the topics I leave below:

Response:

We thank you for reviewing our manuscript. We are glad that you understand the significance of this work. Please find our point-by-point responses to your comments below. The revisions are highlighted in yellow in the revised manuscript.

#1 The title is quite informal, particularly following the use of "sort out". Authors to consider changing the title.

Response:

Thank you for pointing this out. We have revised the title to a more formal text, which is as follows: “Anti-Mycobacterial Drug Resistance in Japan: How to Approach This Problem?”

#2 The abstract does not specify the aims of the review, why it brings novelty to the current literature, how the review is organised and conclusions. Authors to consider reviewing the abstract so key information is added.

Response:

Thank you for your comment. We have discussed the revision of the abstract among the authors and added some sentences to address the problem the reviewer raised. (Line 15–17, Line 34–36) We hope it will make sense.

#3 Chapter 1, the introduction, has little information on what is happening in Japan. If the centre of the review is the anti-TB therapies in Japan, however there is not enough introductory facts from Japan that makes important to review and discuss data. Authors to consider adding more data from Japan, in order to accurately introduce the review.

Response:

We sincerely thank you for your comment. We have added a comment on the trend of DR-TB in Japan.(Line 113–114) Additionally, the drug resistance of TB is fully discussed in section 3.1.1.

#4 Chapter 2, on drug susceptibility testing and limitations. Again, no data from its use in Japan. Authors to consider adding more data from Japan, in order to accurately address the reality and challenging of all the techniques in Japan. In other words, it would have to be clear why the authors develop this chapter so well in a review around TB in Japan.

Response:

Thank you for your comment. Section 2 discusses DST and its limitations in general terms, not limited to Japan, for both TB and NTM. The reason for this is that, for many readers, these issues are not common information and need to be explained in detail. Regarding the issues specific to Japan, as you have pointed out, there was a lack of information on MAC; therefore, we created a new section, i.e., section 3.2.3, and discussed the issues (lines 510–520). TB and MABS are described in sections 3.1.1 and 3.3.2, respectively, and part of them has been revised (lines 545–547).

#5 Chapter 3 should be somehow linked with the chapter 1 and 2. In my opinion, it would be interesting to start with overall facts prior to a focus on Japan, for all the subchapters. Authors to consider rewriting the review, including facts about Japan throughout the paper, and as an isolated chapter. It makes the overall manuscript more interesting for the reader, as facts between countries, for example, can be compared. The authors can provide factual comparisons.

Response:

Thank you for your opinion. As mentioned above, section 2 describes the overall facts and section 3 focuses on Japan.

#6 Subchapter 3.3.2. - Are the authors able to provide references for this paragraph?

Response:

Thank you for the comment. There is no large scale study with BrothMIC NTM for RGM, because the manufacturer’s instruction clearly mentions the kit is not for RGM. Then, case reports of RGM infection in Japan with inappropriate DST are cited as references 64, 65.

#7 Chapter 4 - This is a chapter that could be generalised for all countries and not just Japan. In my opinion it this chapter would be interesting if the authors would compare drugs not available in Japan but available/recommended by the FDA, EMA, NICE/NHS, WHO, in order to "present" or suggest the future that Japan can have.

Response:

Thank you for your valuable opinion. It is certainly a problem that there are several drugs that cannot be used in Japan, but the existence of these drugs does not really help solve the fundamental problem. Rather, we believe that it will be more beneficial to conduct new research by highlighting and informing the public about potential drugs. We have added this opinion to chapter 4 (lines 553–557).

#8 Chapter 4 - Can the authors justify why pretomanid is not in this list?

Response:

Thank you for your comment. Although pretomanid is one of the combination drugs, it has already been shown to be highly effective against MDR-TB in BPaL; hence, it was not included in section 4.

Moreover, pretomanid use is not approved in Japan, and the same class of delamanid that is available is often included in the treatment regimen in MDR-TB. This information has been added to section 3.1.1 (lines 438–442)

Round 2

Reviewer 3 Report

Dear authors, please see my notes below, organised by the different points raised before:

1 - Perfect intervention.

2 - Corrections brought some coherence to the abstract.

3 - Correction appropriately done.

4 - Corrections sufficiently discussed.

5 - NA.

6 - Review discussed appropriately.

7 - Very good discussion added.

8 - This is very good. Thank you.